# Risk Factors and Prevention of Gastric Cancer Development—What Do We Know and What Can We Do?

Paulina Helisz [1] , Weronika Gwioździk [1] , Karolina Krupa-Kotara [1,*] , Mateusz Grajek [2] , Joanna Głogowska-Ligus [1] and Jerzy Słowiński [1]

1 Department of Epidemiology, Faculty of Health Sciences in Bytom, Medical University of Silesia in Katowice, 41-902 Bytom, Poland
2 Department of Public Health, Faculty of Health Sciences in Bytom, Medical University of Silesia in Katowice, 41-902 Bytom, Poland
* Correspondence: kkrupa@sum.edu.pl

**Simple Summary:** There is a wealth of scientific publications in the academic field related to gastric cancer (GC), but the search for methods of prevention is still needed. This paper presents validated findings on the relationship between the microbiota and gastric cancer, indicating the need for further research toward the use of targeted probiotic therapy in the prevention and treatment of GC. Additionally, the key components of an immunomodulatory diet, which may be an important factor in cancer prevention, are presented, and GC risk factors are highlighted.

**Abstract:** Gastric cancer (GC) is one of the most common causes of cancer-related deaths. Gastric tumors show a high aggressiveness, which, in turn, contributes to a low survival rate of fewer than 12 months. Considering the above, it was decided to review the current scientific studies that indicate the potential prevention of gastric cancer and clarify the relationship between gastric cancer and the composition of the microorganisms inhabiting the human body. Accordingly, a review paper was prepared based on 97 scientific sources from 2011 to 2022. Particular attention was paid to the most recent scientific studies from the last five years, which account for more than 80% of the cited sources. Taking care of one's overall health, including undertaking treatment for *Helicobacter pylori* infection, and following a diet high in anti-inflammatory and immunomodulatory ingredients are the most important factors in reducing the risk of developing gastric cancer.

**Keywords:** gastric cancer; nutraceuticals; cancer; microbiota

## 1. Introduction

Gastric cancer (GC) continues to be a major public health problem, especially in terms of mortality. Currently, GC ranks fifth in terms of incidence and is the third cause of cancer deaths worldwide. It seems worth noting, however, that the incidence of GC shows a geographic dependence. The highest rates are found in China, Japan, and Korea, while to a lesser extent in Western European countries, Australia, or North America. Gastric tumors show a high aggressiveness, which, in turn, contributes to a low survival rate of fewer than 12 months [1–3], so the purpose of this review was to compile the latest scientific reports on risk factors for the development of GC and, thus, identify possible preventive measures. To achieve this goal, the following research questions were posed:

Q1: Are there correlations between microbiota status and the development of gastric cancer?
Q2: Can probiotic therapy have a positive impact on gastric cancer prevention?
Q3: Can interventions be taken to reduce the risk of gastric cancer?

## 2. Materials and Methods

### 2.1. Methodology Background

This review aimed to collate the latest scientific reports on risk factors for the development of GC, thereby identifying possible preventive measures. Current scientific research points to gastric cancer risk factors as largely dependent on lifestyle and healthcare. It seems important to identify specific preventive measures. Therefore, the scientific evidence was reviewed based on the available literature.

### 2.2. Review Procedure and Search Strategy

The following study has been carefully redacted based on good practices that are commonly used in papers of this type. The authors of the paper started by defining the research field. To do this, they searched the PubMed database and found scientific articles in line with the topic under consideration. The references were searched by the authors of the paper and a qualified library staff member, using relevant keywords with Boolean operators and their combinations and configurations: gastric cancer, *Helicobacter pylori*, gastric microbiota, gut microbiota, probiotic therapy, and gastric cancer prevention, using the PubMed database as a methodological tool.

### 2.3. Sources Selection

The literature search yielded many records, from which 1789 sources directly related to the topic of the paper were selected, and then those with the highest scientific value were selected according to bibliometric impact factors. The final literature review was based on 97 sources, representing mainly scientific output from recent years (Figure 1).

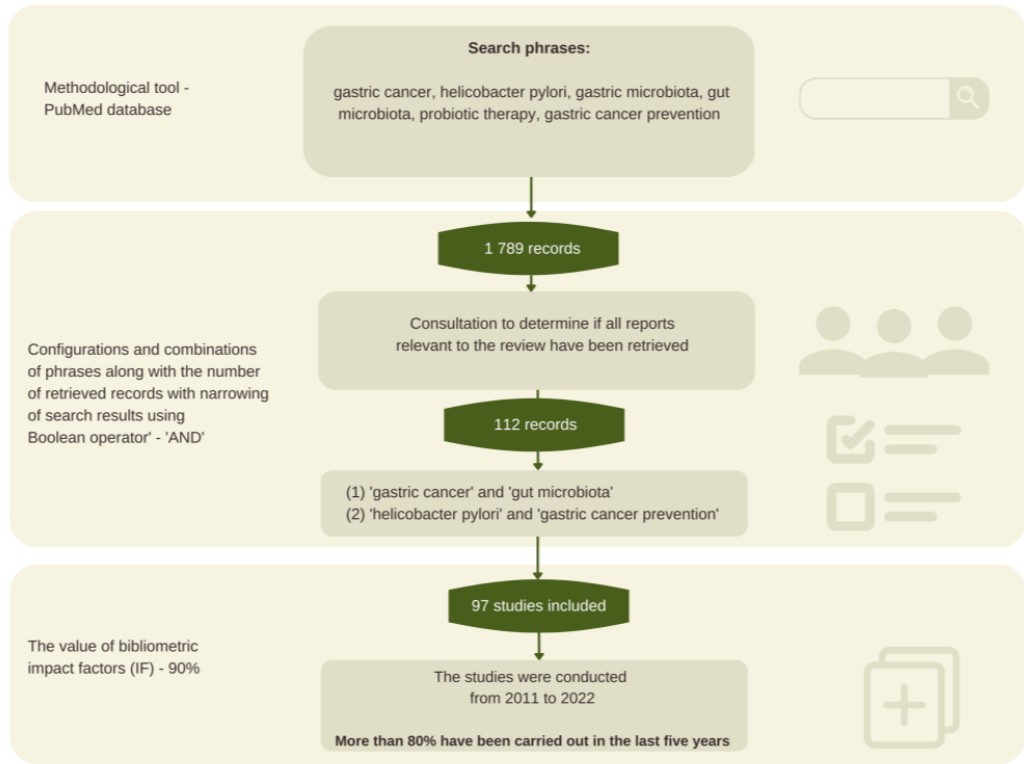

**Figure 1.** Methodological scheme. *Source: own study*.

The reliability, accuracy, and relevance of the work were assessed using the GRADE (The Grading of Recommendations Assessment, Development, and Evaluation) system, one of the main goals of which is to eliminate confusion arising from the use of different evaluation methods. As a result, an overview paper was prepared based on 86 scientific sources from 2011 to 2022. Special attention was paid to the most recent scientific studies of the last five years, which account for more than 80% of the cited sources.

## 3. Gastric Cancer—Characteristics and General Classification

The stomach performs key functions in the human body, ensuring, among other things, proper digestion of food. The organ in question assumes the shape of a pouch, which is why early cancerous lesions do not produce characteristic symptoms, resulting in the majority of GC cases being detected at its advanced stage [2,3]. The process of carcinogenesis is influenced by many genetic and lifestyle-dependent factors. Gastric cancer is characterized by multi-year and multi-stage development and progression. It is estimated that the first cancerous lesions appear approximately 20 years after exposure to carcinogens [3].

To date, gastric tumors have been divided based on their location—cardiac and distal. The former refers to the small paracardial area, while the latter refers to the rest of it. The most popular classification in terms of histology is the Laurén classification, which distinguishes between two types of GC—intestinal and diffuse. The first GC subtype is often associated with *Helicobacter Pylori* and lifestyle, which includes a high intake of table salt and alcohol, a low supply of fruits and vegetables, or smoking. It is estimated that about 15–20% of tumors do not fall under Laurén's classification and are, therefore, considered intermediate tumors [1–3]. Clinically, GC can also be divided according to its early or advanced stage. Early gastric cancers refer to small tumors (2–5 cm) that take the form of invasive carcinoma of the gastric mucosa or submucosa. Detection of lesions at their early stage is associated with relatively good survival [4].

### 3.1. Helicobacter Pylori as a Carcinogen

A constant object of research by scientists is the link between bacterial infections and cancer. Although the topic is quite controversial, at this point, it is known that *Helicobacter pylori* promote the development of GC. The cancer process is a multi-year, multi-stage process that most often begins with chronic gastritis. The vast majority of GC cases are associated with *Helicobacter pylori* infection (about 60%) [5,6], which is classified as a Gram-negative bacterium, and its natural site of occurrence is the surface of gastric mucosal epithelial cells. Infection with the bacterium of the *Helicobacteraceae* family is believed to occur as early as childhood, which explains the reduced ability of gastric mucosal lining cells to produce hydrochloric acid with age. Moreover, it is estimated that more than 50% of people worldwide are infected. Undoubtedly, a peculiar feature of *Helicobacter pylori* is its ability to colonize the stomach for decades, as well as its ability to provide appropriate adaptive conditions surrounding the pathogen by producing large amounts of urease. This enzyme contributes to the degradation of urea to carbon dioxide and ammonia, which, in turn, increases the pH in the stomach due to a previous reduction in hydrochloric acid secretion [5–8].

The International Agency for Research on Cancer (IARC) and the World Health Organization (WHO) have recognized *Helicobacter pylori* infection as an important type I carcinogen in the development of GC, but the pathomechanism of the bacterium's effect on gastric cancer progression is not fully understood, due to the multilevel impact of the pathogen. However, it is known that *Helicobacter pylori* is associated with superficial gastritis and associated inflammation, and this, in turn, promotes a predisposition to gastric ulcers and adenocarcinoma. The literature data indicate that the bacterium of the genus *Helicobacteraceae* contributes to structural changes in the gastric epithelium, thereby causing abnormalities in its function. The above course appears to be crucial in the initial stages of carcinogenesis. The process of progression of gastric and duodenal ulceration is initiated by inflammation caused by *Helicobacter pylori* followed by a decrease in the number of D

cells responsible for the production of somatostatin, which, in turn, acts as a somatoliberin antagonist. As a result, there is an increased secretion of hydrochloric acid, which is induced by the hormone produced by G-cells in the gastric body—gastrin [6–9].

This is because it is emphasized that gastric cancer of the intestinal type is the domain of people of the African race. In contrast, GC with the proximal location of the tumor is the domain of people of the Caucasian race. In addition, it has been proven that environmental factors, as well as lifestyle, play key roles. *Helicobacter pylori* is an important factor in increasing the risk of developing GC, and a meta-analysis conducted by Hooi et al. in 2017 [10] stresses that the highest incidence of the carcinogenic pathogen is found in Africa, Latin America, the Caribbean, and Asia. The lowest incidence, on the other hand, was attributed to North America and Oceania. The reasons for the above geographical distribution in terms of *Helicobacter pylori* incidence are attributed to socioeconomic factors, such as the level of urbanization, access to clean water, sanitation, professional education, and environmental awareness. Analyses of the prevalence of infection with the pathogenic bacterium should take into account the fact that populations are sometimes mixed, which can affect the statistics. Nevertheless, *Helicobacter pylori* infection is often transmitted via the fecal-oral route, so in developing countries without access to clean water, failure to practice proper hygiene, as well as limited access to medical care, may contribute to a higher incidence of infection compared to developed countries [10,11].

*3.2. Gastric Microbiota*

Scientific reports relating to the presence of *Helicobacter pylori* in the gastrointestinal tract have sparked greater interest in the stomach microbiota as a so-called "ecological niche." It turns out that the stomach environment is populated by bacteria with a high degree of diversity, and their density is estimated at $10^1$–$10^3$ CFU (colony-forming units). Studies of the gastric microbiota are inconclusive. In terms of GC development, the potential role of *Lactobacillus* and *Lactococcus* bacteria is highlighted. The theoretical causal relationship is seen in their fermentation end product, lactic acid. Most likely, lactate may contribute to tumor progression and enhance angiogenesis. In addition to *Lactobacillus* and *Lactococcus* bacteria, attention is also drawn to *Nitrospirae* bacteria, which are observed in people with GC. There is no doubt that an inadequate diet plays a key role in cancer progression. Thus, in the case of GC, the carcinogenic importance of nitrates, ubiquitous in smoked and cured products—mainly meat—which can consequently lead to the synthesis of N-nitroso compounds, is emphasized. Bacteria of the *Nitrospirae* type are involved in the metabolic transformation of nitrates and nitrites, so attention is drawn to the carcinogenic role of N-nitrosamines produced by the described pathogen, which may explain their presence in the stomach of GC patients [4,8,12,13].

On the other hand, it is emphasized that chronic *Helicobacter pylori* infection provides a friendly environment for the growth of new bacteria, through reduced secretion of hydrochloric acid. A study conducted on mice clearly shows that *Helicobacter pylori*-infected rodents with concomitant hypergastrinemia have a more complex gastric microbiota in GC compared to healthy animals. A study by Ferreira et al. [9] showed that patients with chronic gastritis were characterized by increased amounts of bacteria such as *Helicobacter*, *Streptococcus*, *Prevontella*, and *Neisseria*, regardless of the presence of *Helicobacter pylori* infection. In contrast, *Citrobacter, Clostridium*, and *Lactobacillus* were observed in patients with achlorhydria and GC. In addition, the authors of the study highlighted the role of *Citrobacter rodentium* in the process of epithelial cell proliferation, which, in turn, promotes the development of colon tumors in rodents with a genetic load for CC [9]. A 2016 study by Wang et al. [13], on the other hand, found an increased abundance of *Escherichia-Shigella* in GC patients. The researchers also highlighted the potential genotoxic effects of *E. coli* in the development of gastric cancer [13].

*3.3. Gut Microbiota and Gastric Cancer*

Each year, the number of published papers in the field of gut microbiota and its impact on the host body increases. According to current knowledge, the microbiota of the lower gastrointestinal tract is the most complex and dynamic ecosystem in the human body. It is well known that intestinal dysbiosis has a positive correlation in the aspect of colorectal cancer [14–16]. At the time of editing this paper, online database search engines displayed access to a few papers relating to the gut microbiota in the aspect of gastric cancer, which may be due to the limitations and analytical difficulties of the microbiome of this section of the gastrointestinal tract. However, there is no doubt that this topic is becoming an increasing object of interest, due to the desire to understand the pathomechanism of the development of malignant neoplasms, which are among the leading causes of death, and to develop chemoprevention methods. Nevertheless, already several years ago, there were the first firm claims that changes in the intestinal microbiome in the gastrointestinal tract may have a significant impact on the development of gastric cancer [17]. It has also been pointed out that a higher abundance of *Enterobacteriaceae* is associated with all types of gastric cancer, which could potentially be useful as a marker of gastric cancer [18]. It is also worth noting the mechanism by which the microbiota influences carcinogenesis. Chronic inflammation and infections are important causes of gastric cancer development due to carcinogenic mechanisms, such as inducing mutations and altering gene status, promoting angiogenesis and cell proliferation [19]. The dysbiosis microbiota and its metabolites affect carcinogenesis not only by inducing inflammation and immune dysregulation, leading to genetic instability but also by interfering with drugs used in anticancer therapy [20].

One effective method of GC prevention is *Helicobacter pylori* eradication. Unfortunately, its adverse consequences include an impact on the disruption of the gut microbiota. Individuals with a history of *Helicobacter pylori* infection showed a decreased abundance of *Bacteroidetes* and increased *Firmicutes* and *Proteobacteria* [14]. In contrast, those infected with *Helicobacter pylori* and simultaneously suffering from the aggressive gastric disease had markedly lower abundances of several *Bifidobacterium* species in the lower intestine, indicating a reduced abundance of protective bacteria [21]. Another study noted an association between *Helicobacter pylori* infection and the abundance of bacteria from the genera *Enterococcus*, *Lachnoclostridium*, *Tyzzerella*, *Roseburia*, *Butyricicoccus*, *Dorea*, *Halomonas*, and *Burkholderiales*. Moreover, an association has been observed between intestinal dysbiosis and the incidence of obesity and older age [22]. In turn, other sources indicate that successful eradication of *Helicobacter pylori* potentially restores the gastric microbiota to a state similar to that found in uninfected individuals and has been shown to have beneficial effects on the gut microbiota [23].

Clinical studies have noted an altered gut microbiota during the perioperative period. In the intestinal microbiota of GC patients, a higher abundance of *Escherichia/Shigella*, *Veillonella*, and *Clostridium* XVIII and a lower abundance of *Bacteroides* were observed compared to healthy controls. In contrast, the abundances of *Akkermansia, Escherichia/Shigella*, *Lactobacillus*, and *Dialister* genera changed after surgery [24]. This indicates the problem of intestinal dysbiosis not only as a potential risk factor for the development of gastric cancer but also as a consequence of its occurrence, which affects the patient's quality of life and health consequences. In addition, it is noteworthy that the choice of treatment has an impact on the gut microbiota of patients with gastric cancer—the surgery will result in a significant change in the gut microbiota [25].

Commonly used proton pump inhibitors (PPIs) in the course of *H. pylori* infection bear their consequences on the patient's gut microbiota. The effect of PPIs is to raise the pH of the stomach, which has been disturbed via a pathogen of the *Helicobacteraceae* genus. Proper use of PPIs, as prescribed by the doctor, brings the expected therapeutic results. However, it is worth noting that long-term intake of proton pump inhibitors contributes to the development of intestinal disorders such as small intestinal bacterial overgrowth (SIBO) and *C. difficile* infections. PPIs are widely available over-the-counter drugs, so statistics on the use of the aforementioned pharmaceuticals may be underestimated. The effect of the discussed group of drugs does not refer only to increasing the pH in the stomach but also to blocking the action of the so-called proton pump or hydrogen-potassium ATPase, which, in turn, contributes to hypochlorhydria, which explains the increased risk of *C. difficile* infection and SIBO [26,27]. The prevalence of small intestinal bacterial overgrowth among gastric cancer patients is confirmed by clinical studies [24,26–28]. A scientific article edited by Liang et al. [24] highlights the prevalence of SIBO in patients with gastric and colorectal malignancies, as nearly 70% of the study's participants, claiming to have taken PPIs for a long time, struggled with the intestinal condition addressed [24].

## 4. Probiotics

According to the WHO definition, probiotics are called "live microorganisms that, when administered in adequate amounts, produce health benefits in the host body." Probiotic bacteria, when used according to the manufacturer's recommendations, can positively affect a disturbed intestinal microbiota by normalizing it. Depending on the type of bacterial strain included in the product, its probiotic effects can vary [26].

Depending on the stage of GC and the location of the primary tumor, different methods of cancer treatment are undertaken. Surgical removal of the stomach (gastrectomy) is one of the most effective treatment strategies used in oncology, especially in the course of GC. A total or partial gastrectomy is associated with several postoperative problems, where the main focus is on the high risk of malnutrition and a significant decrease in immunity [29,30]. Disturbed composition of the gut microbiota (dysbiosis) is associated with dysregulation of the immune system, which is why Zheng et al. in 2019 [30] conducted a study on a group of patients with gastric cancer after subtotal gastrectomy, who were given a diet with a probiotic (a combination of *Bifidobacterium infantis*, *Lactobacillus acidophilus*, *Enterococcus feacali*s, and *Bacillus cereus*) after 3–5 days. The cited 2019 study showed that the above combination of probiotics positively correlated with increased immunity in patients, based on blood tests, specifically lymphocytes, albumin, and total protein. In addition, biochemical studies showed that the use of a combination of *B.infantis, L. acidophilus, E.feacalis*, and *B.cereus* positively reduced inflammation in the body of patients undergoing partial gastrectomy [30]. Similar results were obtained by Xie et al. in 2018 [31], where patients with GC after distal gastrectomy were placed with a nasogastric probe to start early enteral feeding (EEN) as soon as possible. The study group received probiotics combined with EEN, while the control group received EEN alone. The cited 2018 study showed significant differences in IgG, IgA, and IgM levels between the study group and the control group. In addition, it was proven that the combination of probiotics with EEN had a clear effect on reducing the inflammatory response (IL-6, IL-8, and TNF-α) and the incidence of diarrhea among the study patients [31]. Concordant results were obtained in a meta-analysis conducted by Yang et al. [19], which emphasizes the positive effect of probiotics on GC patients. Adequate supply probiotics is associated with a reduced incidence of side-effects that accompany the course of the disease, as well as following the associated treatment (mainly surgical). The most common symptoms include diarrhea, constipation, bloating, and vomiting, but also intestinal dysbiosis, and urinary tract and abdominal infections. In addition, an analysis by Yang et al. also found that probiotics help reduce levels of IL-6, TNF-α, and CRP, among others, and thus inflammation [19].

Gastrointestinal cancer patients have a high risk of developing SIBO. A study by Liang et al. found that a group of patients who received *Bifidobacterium* supplementation for 4 weeks showed more than half the incidence of small intestinal bacterial overgrowth syndrome compared to the placebo group [24]. A lot of hope for the highly positive effects of probiotics on the immune system, among other things, is seen in the bacterial strains *Bifidobacterium* spp. and *Lactobacillus* spp. The relationship between the effects of the above types of microorganisms on local and systemic effects on the host body is currently being sought by scientists around the world. It is worth mentioning that reliably conducted clinical trials could significantly improve the effectiveness of the treatment of oncological patients, especially patients with gastrointestinal cancers [32].

## 5. Nutraceuticals with Chemopreventive Effects

### 5.1. Dietary Fiber

Dietary fiber belongs to the carbohydrate group and has been credited with many health benefits since the 1970s. Dietary fiber is generally defined as oligosaccharides (fructo- and galactooligosaccharides), non-starch polysaccharides (lignin, hemicellulose cellulose, pectin, and hydrocolloids—including β-glucans), and resistant starch. The effect of dietary fiber is a constant object of research by scientists all over the world; however, according to current knowledge, it is known that the fiber in question shows a protective effect against cancer (mainly colon cancer), but also cardiovascular diseases. In addition, dietary fiber contributes to the proper modulation of the intestinal microbiota, thus preventing the development of inflammation caused by dysbiosis [33,34].

In the context of anticancer activity, fiber is believed to be a component that counteracts, among other things, the carcinogenic effect of nitrosamines, produced mainly during the heat treatment of food [35]. There are still no conclusive studies indicating the efficacy of increased dietary fiber intake against lowering the risk of gastric cancer, but there are indications suggesting benefits from an increased supply of this component [35,36]. One study found that increasing fiber intake by 10 g/day resulted in a 44% reduction in gastric cancer risk [37]. A study conducted in Korea that analyzed gastric cancer risk about dietary patterns showed that a diet high in fruits and vegetables, and, therefore, a source of dietary fiber, correlated positively with a reduced risk of gastric cancer [38]. In contrast, another study of 91,946 Japanese at a 15-year follow-up found that dietary fiber intake was not associated with gastric cancer risk [39].

Despite the lack of conclusive evidence for the efficacy of increased dietary fiber intake in the context of preventing the development of GC, it is worth bearing in mind the effect of fiber on the gut microbiota, which has a clear link to the condition. Dietary fiber is the most popular substance that exhibits prebiotic effects. Prebiotics are undigested food components that show a stimulating effect in terms of the growth of beneficial bacterial strains [26,40,41]. Fiber function refers to the process of hydrolysis by digestive enzymes present in the small intestine, as well as fermentation by intestinal bacteria in the colon. It seems noteworthy that dietary fiber (e.g., inulin) added to foods improves the sensory properties and texture of foods. In product studies, it has been shown that the addition of prebiotic substances to yogurts was associated with higher evaluations by consumers of the taste and other organoleptic qualities of the products analyzed [41]. In addition, dietary fiber is an important component for maintaining the continuity of mucous membranes and, therefore, has an indirect immunoprotective effect [42].

### 5.2. Polyunsaturated Fatty Acids

In the context of anticancer activity, omega-3 fatty acids are particularly important in the group of polyunsaturated fatty acids [43]. Available studies indicate that the ratio of omega-3 to omega-6 fatty acids is important in reducing cancer risk, and this is related to their different mechanisms of action. It has been shown that omega-3 fatty acids can inhibit cell viability by inducing apoptosis and affecting the expression of genes associated with tumorigenesis, suggesting their use as therapeutics against gastric cancer [43]. Moreover, during carcinogenesis, it is important to change the composition of the dietary membrane, due to the increased rate of lipid synthesis in tumor tissues. The incorporation of omega-3 fatty acids into the cell membrane regulates inappropriate cell proliferation and reduces inflammation-related carcinogenesis [44]. This is important not only in the course of cancer but also in prevention [45]. In addition, the prevention of malnutrition is extremely important during therapy. An adequate supply of omega-3 fatty acids has shown promise in combating cancer cachexia by preventing weight loss and increasing lean body mass, as well as reducing inflammation and maintaining the immune profile of patients [46,47]. In contrast, another study found that the addition of omega-3 fish oil fat emulsion to parenteral nutrition after laparoscopic surgery in patients with gastric cancer could effectively reduce the incidence of the onset of short-term postoperative complications and, thus, have a positive impact on the recovery process [48]. Omega-3 fatty acids also have the potential to increase the effectiveness of chemotherapy and alleviate gastrointestinal symptoms. Nausea, vomiting, as well as bowel problems (diarrhea/constipation), are commonly observed in patients treated with chemotherapy [49]. The above symptoms are most common in patients with colorectal and gastric cancer. Cytostatics used in oncology affect damage to healthy rapidly dividing cells, resulting in irritation, atrophy, and pain associated with changes in mucous membranes, such as the patient's mouth. In addition, aforementioned diarrhea and constipation may be the result of changes in the intestinal microbiota caused by cytostatic treatment. Scientific studies show that EPA and DHA fatty acids have a beneficial effect on the restoration of cell membranes [49,50]. Furthermore, the importance of fish oil for intestinal disorders is highlighted. In addition, in the group of patients who were supplemented with fatty acids, a weaker intensity of diarrheal episodes was observed compared to the control group (placebo) [51]. Because the human body synthesizes EPA and DHA to a limited extent, supplementation and adequate intake of products rich in the above-mentioned acids seem to be essential [50]. Table 1 shows selected foods high in EPA and DHA.

**Table 1.** Selected products with high DHA and EPA content [52,53].

| Food Product | FA Content (g/100 g of Product) | EPA [1] (mg/100 g FA [3]) | EPA (mg/100 g of Product) | DHA [2] (mg/100 g FA [3]) | DHA (mg/100 g of Product) |
|---|---|---|---|---|---|
| Mussels, cooked | 2.20 | 22,350 | 491.7 | 8350 | 183.70 |
| Prawns, cooked | 0.90 | 18,250 | 164.25 | 14,110 | 126.99 |
| Cod, baked | 0.50 | 8290 | 41.45 | 25,700 | 128.50 |
| Mackerel, grilled | 22.40 | 5820 | 1303.68 | 9160 | 2051.84 |
| Haddock, steamed | 0.60 | 11,260 | 67.56 | 29,740 | 178.44 |
| Chicken, breast, grilled without skin | 2.2 | 470 | 10.34 | 1010 | 22.22 |
| Tuna, canned in sunflower oil, drained | 6.4 | 400 | 25.6 | 2250 | 144 |

**Table 1.** *Cont.*

| Food Product | FA Content (g/100 g of Product) | EPA [1] (mg/100 g FA [3]) | EPA (mg/100 g of Product) | DHA [2] (mg/100 g FA [3]) | DHA (mg/100 g of Product) |
|---|---|---|---|---|---|
| Salmon, pink, canned in brine, drained | 4.8 | 8000 | 384 | 14,650 | 703.2 |
| **RecommendedDHA + EPAintakeforadults: 250 mg perday** | | | | | |

[1] EPA—eicosapentaenoic acid, [2] DHA—docosahexaenoic acid [3] FA—fatty acids.

In the aspect of gastric cancer prevention, omega-3 fatty acids have their uses, potentially blocking carcinogenesis processes associated with *Helicobacter pylori* infection. The presumed protective effect is attributed to the antimicrobial activity of the n-3 fatty acids in question, which, in turn, contributes to the inhibition of colonization of the gastric mucosa by the carcinogenic pathogen [54]. The opposite effect is demonstrated by cholesterol, which can be transformed by the *Helicobacter pylori* gene, thus enabling it to increase its resistance to administered antibiotics, including the commonly used in eradication therapy—amoxicillin and clarithromycin [55]. One study by Correia et al. showed that providing DHA acids at 250 μM during antibiotic therapy, in *Helicobacter pylori*-infected mice, effectively prevented the recurrence of reinfection. It seems noteworthy that less than 100 μM of DHA supply provided an inhibitory effect on the growth aspect of the carcinogenic pathogen [56]. Similar results were obtained by Han et al. who observed a markedly lower expression of proteins involved in cellular proliferation processes in mice. In addition, the authors of the study emphasized that the long-term effect of ω-3 PUFA use may contribute to the inhibition of gastric carcinogenesis processes caused by *Helicobacter pylori* infection [57]. In contrast, Javid et al.'s study highlights the role of omega-3 fatty acids as one of many factors that increase the occurrence of oxidative stress in the body, which is the result of the reduced expression of antioxidant enzymes. The authors of the paper emphasized that the treatment of *Helicobacter pylori* with concomitant supplementation of n-3 fatty acids in combination with antioxidants may show greater efficacy than omega-3 supply alone [58].

The Western diet is characterized by an inadequate ratio of omega 3 to omega 6 fatty acids. A high intake of products rich in n-6 and a low intake of n-3 PUFAs predispose to the development of diseases of various causes, including cancer or cardiovascular disease, the leading cause of death worldwide [59]. What is extremely important is the ratio of omega-3 fatty acids to the omega-6 family, which is a determinant of the health-promoting properties of the diet. According to recommendations for proper nutrition, it should be 1:(4–5). An excessive supply of omega-6 fatty acids can lead to a reduction in the beneficial biological effects of omega-3 fatty acids. However, the current common diet is characterized by an imbalance in the ratio of fatty acids consumed and an excessive intake of saturated fatty acids, as well as too little intake of omega-3 fatty acids while consuming too much omega-6 fatty acids [60]. An adequate supply of omega-6 fatty acids also shows its beneficial effects. In addition, arachidonic acid, commonly found in animal products such as eggs, meat, and offal, is one of the most important substrates in the metabolic pathway. Moreover, it can account for up to 25% of all fatty acids present in skeletal muscle or immune cells. Arachidonic acid plays an important role in maintaining the body's homeostasis and regulating inflammatory and immune processes. The n-6 group acid in question, such as docosahexaenoic acid, also has a cognitive and developmental function in the brain and is an important element in the bioactivity of lipid mediators. Linoleic acid, on the other hand, present in products of plant origin (mainly oils and nuts), is an essential fatty acid that ensures proper growth and development of the body [61]. In addition, linoleic acid has important functions at the cellular level; its deficiency can affect the abnormal activation of immune cells and affect disorders related to cell–cell adhesion [62]. It has been suggested

that an appropriate ratio of n-3 to n-6 PUFA acids is one of the strategies to reduce the risk of cancer, and is, therefore, one of the preventive methods [59].

### 5.3. Ingredients with Immunomodulatory Effects

An immune-modulating diet is an important element in the prevention of both gastric cancer and other cancers. The issue of the body's immunity has been an important object of interest for scientists for many years, but with the development of the SARS-CoV-2 pandemic, it has gained additional importance. There is a constant search for an ideal immunomodulating dietary pattern. Recently, the POLA index, which evaluates the immunomodulatory potential of a diet, was created [62]. The authors suggest using this tool to identify individuals who are eating a diet that is deficient in components that support the immune response, thereby changing their dietary behavior. To assess the immunomodulatory nature of the diet, the supply of components with proven effects in this regard was considered, including iron, zinc, calcium, folate, linoleic acid, α-linolenic acid, vitamins A, D, E, B1, B6, and C, and fiber [62]. Table 2 shows the immunomodulatory effects of selected nutrients.

**Table 2.** Immunomodulatory effect of selected nutrients.

| Nutrient | Immune Function |
|---|---|
| Iron | It likely influences B-cell function and Th1/Th2 lymphocyte balance [63]. Affects the intestinal microbiota, by influencing the growth and survival of microorganisms inhabiting the human body [64]. Increased iron most likely promotes intestinal inflammation, while low Fe levels positively correlate with intestinal infections [64,65]. Potential anticancer effects through immune modulation and a hypothetical component of supporting cancer treatment in terms of tumor suppression (importance of ferroptosis) [66]. |
| Zinc | Zinc deficiencies contribute to increased production of pro-inflammatory cytokines such as interleukin-1β, interleukin-6, and TNF-α [67,68]. Deficiency results in decreased numbers of granulocytes, lymphocytes, and NK cells [67–69]. An essential micronutrient in both acquired and innate immunity processes [68,69]. Anti-inflammatory activity via regulatory T cell (Treg) function and by inhibiting NF-κB (nuclear factor κ-light-chain-enhancer of activated B cells) [69]. Exhibits indirect anti-tumor activity mediated by Th17 cells, NK cells, and T cells (Treg) [67]. |
| Calcium | It contributes to cellular functions such as proliferation, apoptosis, and gene transcription [70]. Calcium ions contribute to the proper functioning of cytotoxic T lymphocytes and NK cells, which may be important in terms of anticancer therapies [71]. |
| Dietary fiber | The function of pectins includes strengthening the mucus layer, increasing epithelial integrity, and activating or inhibiting dendritic cell and macrophage responses. Pectins can strengthen the intestinal immune barrier by promoting the adhesion of commensal bacteria and inhibiting the adhesion of pathogens to epithelial cells [72]. A high intake of dietary fiber leads to an increase in the number of intestinal bacterial species responsible for the production of SCFAs, essential for the proper functioning of the immune system and inflammatory disease prevention [73]. |
| Omega-3 fatty acids | It contributes to the activation of immune cells, including by regulating cell membranes [74]. It influences changes in gene expression in macrophages [74]. It contributes to the reduction in inflammation through properties that reduce the secretion of IL-1β, TNF-α, and IL-6 [74]. It most likely contributes to T-cell modulation [74]. |

**Table 2.** *Cont.*

| Nutrient | Immune Function |
|----------|-----------------|
| Vitamin D | It affects the immune system, both specific and non-specific immunity [75]. Exhibits anti-tumor activity, through the regulation of tumorigenesis [76]. Contributes to the regulation of the inflammatory microenvironment, via mechanisms such as inhibition of NF-κB (nuclear factor κ-light-chain-enhancer of activated B cells) pathways, and regulation of immune cell cytokine levels [77]. |
| Vitamin C | It supports epithelial barrier functions against pathogens [78]. Plays a key role in immune cell proliferation and differentiation, among other things, and is a cofactor in gene transcription and immune cell signaling [79]. |

The cancer process is multistage and complex, and the prevailing inflammatory process in the human body promotes tumor growth. The characteristics of tumors refer, among other things, to the fact that they exhibit angiogenesis and can effectively evade detection by the body's defense mechanisms [80]. In terms of resistance, the key element appears to be lactic acid, which is produced by tumor cells and is the end product of metabolic transformation—glycolysis. It is suggested that the synthesis of the compound in question provides the right conditions for the development of oncogenic processes such as angiogenesis, proliferation, or increased tumor resistance to administered drugs. In addition, scientific studies confirm the negative effect of lactic acid on the aspect of immune cells, which affects both acquired and innate immune cell functions [81,82].

The process of apoptosis is among the key functions of the body to ensure the maintenance of homeostasis. Programmed cell death contributes to the balance between cell gain and loss. In the case of tumorigenesis, the process of apoptosis is impaired, and the proliferation of mutant cells is much higher than their life cycle [83]. In the aspect of regulating cell proliferation, vitamin D3, through its receptor, shows its activity in this regard. In addition, cholecalciferol contributes to the induction of apoptosis [84]. Because the cancer process is complex and the pathomechanism of cancer is a constant object of research by scientists around the world, it seems important to take care of the body's immunity as one of the key elements of cancer prevention [85].

Diet plays an important role in the prevention of gastric cancer. A well-balanced diet and, above all, varied nutrition will be key chemopreventive factors in the aspect of GC. However, special attention should be paid to dietary supplements and bioactive substances, as some of them may have the effect of increasing the risk of gastric cancer, such as aristolochic acid, an ingredient in Chinese herbs and weight loss pills [86]. Therefore, dietary supplementation should be approached with caution and limited to the need in relation to nutrient deficiencies or physician recommendations.

A gastric-cancer-risk-reducing diet should primarily be low in processed foods high in fat and sugar [87,88]. The main source of simple carbohydrates in the diet should be products that contain it naturally, such as fruit and dairy products. In contrast, vegetable oils, fish, and products high in unsaturated fatty acids, such as olive oil, avocado, nuts, or flaxseed, should predominate as sources of fat, due to the need to reduce saturated fats in excess used for frying [89]. Regarding protein sources, it is suggested to reduce meat consumption if it is predominantly plant-based products. It is better to consume poultry meat, as red meat is considered to increase the risk of carcinogenesis. Having a varied diet, based on polyphenol-rich plant products, will also have a positive impact on reducing the risk of gastric cancer, as this increases the diversity of the microbiome [90]. It is also important to compose meals appropriately so that they have a low glycemic load. Influences on lowering the glycemic load include fiber and fat, as well as the use of products with a low glycemic index [87]. The preferred form of processing food is boiling, steaming, stewing, or baking, as frequent frying of food is among the factors that increase the risk of gastric cancer [88]. For heat-processed foods, it is important to remember that the temperature of the dish should not be high [87]. It is worthwhile to ensure the immunomodulatory properties of the diet by using foods rich in the ingredients mentioned

in this section. Another important peri-nutritional factor for reducing the risk of gastric cancer is to adhere to the principles of so-called "food hygiene." Eating meals at regular intervals, without snacking in between, has a positive effect on the proper functioning of the migrating motor complex, which is important for the proper composition of the microbiome and improves the digestive process [91]. In addition to maintaining meal regularity, the eating process itself is also important. Chewing food slowly and carefully supports proper digestion and gastric function [92]. It is also worth paying attention to methods of managing stress and avoiding eating meals in a state of increased tension, as this has a negative impact on the functioning of the entire gastrointestinal tract. An equally important part of GC prevention is taking care of the products consumed, i.e., properly washing fruits and vegetables before consumption, and checking expiration dates [93].

## 6. Strengths and Limitations

According to the authors, the most effective method of disease prevention is to take a holistic approach to health and take care of health on many levels simultaneously. There are many scientific publications in the scientific space related to gastric cancer, but the search for prevention methods is still needed. This paper presents confirmed reports on the relationship between gut microbiota and gastric microbiota and gastric cancer, pointing to the need for further research toward the use of targeted probiotic therapy in the prevention and treatment of GC. In addition, we outlined the key components of an immunomodulatory diet, which may be an important factor in cancer prevention, and highlighted GC risk factors, which is undoubtedly a strength of the study.

Wanting to point out various proven prevention options, the authors relied on available scientific studies. The topic of microbiota and its importance in cancer is an area that is constantly evolving, and as a result, there is a steady stream of research on the subject [14,94]. The authors are aware that in the face of such a large number of studies, important reports may have been overlooked, but it should be noted that every effort was made to ensure that this review was conducted fairly, taking into account large, multi-center research projects and highlighting mainstream studies. Basing the selection of articles on bibliometric analysis with the impact factor can lead to publication bias, resulting from the desire to publish "positive observations," from high-scoring journals, which is something we tried to avoid.

It is also important in further reviews and studies to address the topic of linking the etiopathogenesis of gastric cancer to psychological and emotional factors and the psychosocial functioning of patients living with this cancer. As the previous experience of the authors of the minor review shows, an important traumatizing and stressful factor in the oncology population may be the COVID-19 pandemic [95,96], so further research will be based on these topics.

## 7. Conclusions

Early detection of gastric cancer in most cases allows for successful treatment, so prevention and education of the public about it is extremely important. There are correlations between the state of the microbiota and the development of gastric cancer—as a result of dysbiosis, the microbiota and its metabolites can cause inflammation and immune dysregulation. This can lead to genetic changes, as well as affect the effects of anti-cancer drugs. Therefore, probiotic therapy may be a potential means of supporting both the prevention and treatment of gastric cancer, but there is a need for further research in this area. Taking care of overall health, including undertaking treatment for Helicobacter pylori infection, and eating a diet with a high supply of anti-inflammatory and immunomodulatory components are the most important factors in reducing the risk of developing gastric cancer.

**Author Contributions:** Conceptualization, K.K.-K.; methodology, K.K.-K., P.H., and W.G.; formal analysis, K.K-K., M.G., P.H., and W.G.; investigation, K.K.-K., P.H., and W.G.; resources, K.K.-K., M.G., P.H., and W.G.; writing—original draft preparation, K.K.-K., P.H., and W.G.; writing—review and editing, K.K.-K., M.G., P.H., and W.G.; supervision, J.G.-L. and J.S.; project administration, K.K.-K.; funding acquisition, K.K.-K. All authors have read and agreed to the published version of the manuscript.

**Funding:** This research received no external funding.

**Institutional Review Board Statement:** The research complies with the provisions of the Helsinki Declaration.

**Informed Consent Statement:** Not applicable.

**Data Availability Statement:** Not applicable.

**Conflicts of Interest:** The authors declare that they have no conflict of interest.

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
