# Peer review of "Risk Factors and Prevention of Gastric Cancer Development—What Do We Know and What Can We Do?"

_onco, doi:10.3390/onco3010003_

Round 1

Reviewer 1 Report

Thank you for the opportunity to review your article.

This article is a review summarizing current scientific studies that point to the possible prevention of gastric cancer and clarify the relationship between gastric cancer and the composition of microorganisms in the human body.

The methodological part of the article describes in detail the selection of articles and the selection criteria.

Page 3 line 92: - correct the number and the unit.

Is there a consensus on the microbiota diversity associated with GC, in terms of which bacteria are dominant and possibly involved in carcinogenesis?

Table 1. shows the sources of DHA and EPA, but also includes smoked fish. Smoked products increase the risk of GC as shown in lines 136-137 on page 4.

I propose to revise and correct Table 1.

Page 8, lines 330-331- you mentioned “supplementation and adequate intake of products rich in the above-mentioned acids seem to be essential” - What does adequate actually mean?

One of the aims of this paper was to analyse whether probiotic therapy could have a positive effect on the prevention of stomach cancer. However, only the role of probiotics after gastrectomy was described.

Since the microbiota is mainly influenced by diet, there is a lack of dietary recommendations for the prevention of gastric cancer. Table 2 contains only nutrients.

Author Response

Dear Reviewer,
Thank you for your insightful analysis of our masnuscript and your many valuable comments, which we hope allowed us to sufficiently improve our article. We have rewritten Table 1 and added a description of diet in the prevention of gastric cancer (in brown in the text). Unfortunately, at this point, database search engines do not display literature items on probiotic therapy in terms of GC prevention. The role of the gastric microbiota is not fully understood. The authors thank you for reviewing the paper.

Reviewer 2 Report

In this paper the authors have been focused on the prevention of gastric cancer. The paper is interesting but incomplete - some aspects need to be modified or added, as follows:

1. Chapter 3 was titled "Gastric cancer". After Introduction and methods pleasde use other sub-tile - "Gastric cancer" is too general. Moreover, first paragraph (lines 74-81) needs to be removed. Describing the stomach does not add any scientific information. 

2. Gastric cancer - the authors mentioned "diffuse GC is recognizable as sin- 88 gle cells without glandular structures (lines 88-89)." Please ask the pathologist to add a proper definition and rephrase it.

3. The authors mentioned that the 5 year survival for early gastric cancer is 90% - line 94. Please rethink it - such cases might show distant metastases.

4. From lines 95 - The authors mentioned the carcinogenic role of H. pylori - which geographic-related particualrities are described in literature regarding this factor (doi: 10.5114/pjp.2015.54959. )?

5. Gastric microbiota - please add data about aristolochic acid which seems to be involved in gastric carcinogenesis too - especially in Asia.

Minor points:

1. Methods - " helicobacter pylori" - please use italic and highlighted "H" - in fig. 1 do the same

Author Response

Dear Reviewer,
Thank you for your insightful analysis of our masnuscript and your many valuable comments, which we hope allowed us to sufficiently improve our article. 

The reviewer's comments were taken into account. After consultation with a pathologist, this definition was dropped. Survival data were also dropped. The authors changed the heading "Gastric cancer" to be more specific. In addition, a geographic description of H.pylori was added as suggested by the Reviewer (on green in the text). An italic for H.pylori was used. The authors thank you for reviewing the paper.

Round 2

Reviewer 2 Report

The paper quality was improved and can be published in the present form.